# Next Generation Sequencing (NGS) Target Approach for Undiagnosed Dysglycaemia

**DOI:** 10.3390/life13051080

**Published:** 2023-04-24

**Authors:** Concetta Aloi, Alessandro Salina, Francesco Caroli, Renata Bocciardi, Barbara Tappino, Marta Bassi, Nicola Minuto, Giuseppe d’Annunzio, Mohamad Maghnie

**Affiliations:** 1LABSIEM (Laboratory for the Study of Inborn Errors of Metabolism), IRCCS Istituto Giannina Gaslini, 16147 Genoa, Italy; 2UOC Genetica Medica, IRCCS Istituto Giannina Gaslini, 16147 Genoa, Italy; 3Department of Neuroscience, Rehabilitation, Ophthalmology, Genetics, Maternal and Child Health (DINOGMI), University of Genoa, 16100 Genoa, Italy; 4Department of Pediatrics, IRCCS Istituto Giannina Gaslini, 16147 Genoa, Italy

**Keywords:** performance validation, NGS, monogenic diabetes, MODY, Wolfram syndrome, congenital hyperinsulinism

## Abstract

Next-generation sequencing (NGS) has revolutionized the field of genomics and created new opportunities for basic research. We described the strategy for the NGS validation of the “dysglycaemia panel” composed by 44 genes related to glucose metabolism disorders (MODY, Wolfram syndrome) and familial renal glycosuria using Ion AmpliSeq technology combined with Ion-PGM. Anonymized DNA of 32 previously genotyped cases with 33 different variants were used to optimize the methodology. Standard protocol was used to generate the primer design, library, template preparation, and sequencing. Ion Reporter tool was used for data analysis. In all the runs, the mean coverage was over 200×. Twenty-nine out of thirty three variants (96.5%) were detected; four frameshift variants were missed. All point mutations were detected with high sensitivity. We identified three further variants of unknown significance in addition to pathogenic mutations previously identified by Sanger sequencing. The NGS panel allowed us to identify pathogenic variants in multiple genes in a short time. This could help to identify several defects in children and young adults that have to receive the genetic diagnosis necessary for optimal treatment. In order not to lose any pathogenic variants, Sanger sequencing is included in our analytical protocol to avoid missing frameshift variants.

## 1. Introduction

Dysglycaemia refers to an abnormality in blood glucose level, which can include hypoglycaemia (low blood glucose level) or hyperglycaemia (high blood glucose level). Monogenic diabetes is a rare form of inherited diabetes mellitus caused by heterozygous or homozygous defects in a single gene involved in β-cell development or function. It accounts for approximately 1% to 6% of pediatric diabetes patients [1]. In fact, until now, most cases of monogenic diabetes remain undiagnosed [2]. The most common type of monogenic diabetes is MODY (maturity-onset diabetes of the young, MIM # 606391) and, to date, 14 subtypes are known [3] representing 2–5% of diabetes cases in Europe [4,5,6,7]. Permanent neonatal diabetes mellitus (PNDM, MIM #606176) is another form of monogenic diabetes characterized by persistent hyperglycemia within the first 12 months of life in general, requiring continuous treatment. Affected patients often respond well to sulfonylureas, and insulin may not be necessary. Wolfram syndrome (WS, MIM #222300), a rare autosomal recessive neurodegenerative disorder characterized by diabetes insipidus, diabetes mellitus, optic atrophy, and deafness, is caused by bi-allelic defects in WFS1 gene. Patients with familial renal glycosuria (FRG, MIM #233100) have decreased renal tubular reabsorption of glucose from urine in the absence of hyperglycemia and any other signs of tubular dysfunction. It is caused by SLC5a2 mono or bi-allelic mutations. Congenital hyperinsulinism (CHI, MIM #256450) is a rare genetic disorder caused by genetic mutations in several genes that lead to an excess of insulin secretion in pancreatic β-cells. Pathogenic variations in KATP channels are the most common cause.

Genetic testing is strongly recommended for a rapid and accurate diagnosis of monogenic diabetes or CHI in order to allow the clinician to make a correct diagnosis.

In the past 10 years, the next-generation sequencing (NGS) has revolutionized the field of genomics and created new opportunities for basic research [8,9]. NGS enables the development of faster, more comprehensive and cost-effective genetic methods compared to conventional Sanger sequencing.

In this manuscript, we aim to describe the strategy for the validation of a home-made NGS “dysglycaemia panel” consisting of 44 candidate genes related to glucose metabolism disorders (MODY, WS) and FRG, using Ion AmpliSeq technology combined with Ion-PGM. We also reported the results of the first genetic analyses.

## 2. Materials and Methods

### 2.1. Cases Used for Validation

All the procedures adopted for the validation of the home-made NGS “dysglycaemia panel” follow the guidelines reported in ACMG Standards for clinical next-generation sequencing [10].

#### Sequencing

Thirty-two anonymized DNA samples of patients, previously genotyped by Sanger sequencing, were used to optimize the methodology and perform the panel validation. Different types of mutations such as missense, nonsense, frameshift (Figure 1A) localized in genes responsible for monogenic diabetes, WS, and FRG (Figure 1B) were chosen.

### 2.2. NGS Ampliseq Protocol

Genomic DNA was extracted from EDTA whole blood using QIAamp DNA Blood Midi kit (Qiagen GmbH, Hilden, Germany) and purified with Amicon Ultra 0.5 mL (Merk Millipore LTD, IRL) according to standard procedures. Eluted DNA was quantified with Nanodrop Spectrometer (Thermo Fisher Scientific, Waltham, MA, USA).

Primers for 44 genes causative of dysglycaemia and its complications were designed using the Thermo Fisher Scientific Ion AmpliSeq Designer platform (version 5.6; www.ampliseq.com (accessed on 30 March 2023)) according to hg19. The list of the genes included in the NGS panel is reported in Table 1.

The primer pairs allowed the amplification of the exons and their flanking regions of all 44 genes. The resulting panel size was 172.857 kb containing 900 amplicons with a coverage of 99% of the targeted regions. All the uncovered parts of the design were sequenced by Sanger sequencing.

A total of 15 ng of DNA from each sample was combined with the Ampliseq reagents and primer pool for the dysglycaemia according to standard protocol. After this step, the amplicon products were partially digested, and IonExpress Adapters and Barcode sequences were ligated to the library fragments. In the following step, the barcoding libraries were cleaned up using a magnetic bead method (Ion Torrent, Thermo Fisher Scientific, Waltham, MA, USA) and then quantified following the Qubit 2.0 fluorimeter (Thermo Fisher Scientific, Waltham, MA, USA) instructions using HS dsDNA kit. The quantified libraries were pooled and diluted once to 100 pM.

The library was amplified by emulsion PCR on Ion sphere particles (ISPs) using the Ion PGM Hi-Q OT2 Kit according to standard protocol (Life Technologies, Carlsbad, CA, USA). Ion 316 chips were used to sequence eight samples simultaneously. Sequencing was performed on an Ion PGM System (Ion Torrent, Thermo Fisher Scientific, Waltham, MA, USA) using the Ion PGM Hi-Q Sequencing kit according to the manufacturer’s instructions. Sequencing data were analyzed with Coverage Analysis and Variant Caller plugins available within the Ion Torrent Suite software TS 5.18 and contextually with Ion Reporter. All the pathogenic or likely pathogenic variants detected by Ion PGM were confirmed using specific couples of primers by direct sequencing of the PCR products.

## 3. Results

In all the runs, the mean coverage was over 200×, and the mean length of the amplicons was about 200 bp. All data of the variants selected for the evaluation of the “dysglycaemia panel” performance are reported in Table 2, and the results of performance metrics are shown in Table 3. Twenty-nine mutations out of thirty-three were detected by NGS sequencing, while four were missed. Three were frameshift variants ≥ 10bp (c.1279_1358delinsTTACA in GCK exon 10, c.1342_1374del in HNF1a exon 7, c.1261_1280dup19 in SLC5a2 exon 10). One was the duplication of c.872dupC in HNF1a exon 4. Three further variants, classified as variants of uncertain significance (VUS), were detected; all of them were found in the DNA of three unrelated cases in addition to pathogenic variants (Table 4).

## 4. Discussion

In this study, we report the successful validation of a custom-designed targeted Ampliseq panel for the detection of variants causative of dysglycaemia and its complications. Recently, in the literature, several reports on the application of NGS approach for genetic diagnosis of monogenic diabetes have been published [17,18,19]. All of them are designed to allow the detection of genetic defects causative of monogenic diabetes mellitus (i.e., MODY and neonatal diabetes mellitus) and rare diabetes-associated syndromes (i.e., Wolfram syndrome, Alström syndrome, Wolcott–Rallison syndrome, and thiamine-responsive megaloblastic anemia (TRMA)/Roger’s syndrome) [20]. Based on this information, 44 different genes that are commonly mutated in monogenic diabetes mellitus, congenital hyperinsulinism, Wolfram Syndrome, and familial renal glycosuria were been included in our panel design. We selected anonymized subjects previously genotyped in which thirty-three different variants were present: twenty-nine point mutations or frameshift variants causative of deletion/insertion <10 bp and four frameshift variants caused by deletion/insertion ≥10 bp. The molecular test confirms a good ability to detect point and deletion/insertion mutations of less than 10 bp. In fact, 96.5% of the variants were confirmed. Regarding frameshift variants greater than 10 bp, only one out of four (25%) was detected by the NGS test. Sequencing errors remain a major challenge in the single nucleotide analysis using an NGS platform [21,22].

Ion torrent sequencing system performed by PGM measures the hydrogen ions released during the incorporation of dNTP in the amplification of the DNA target template. The protons release causes a decrease of the pH level in the solution present inside the chip. Ph variation is directly proportional to the number of bases incorporated in the template. When target region is rich of homopolymer repeats, the multiple incorporation of dNTP may lead to a high number of hydrogens released, and a high electronic signal was detected by the sensor. Consequently, variants occurring in this genomic region may be missed [23]

In our study, all the four missed frameshift mutations are not located in the genomic region rich of homopolymers on repeat regions. Therefore, we may suppose that the failure of detection was related to the primer design used for the library amplification, even if the theoretical coverage of HNF1a, GCK, and SLC5a2 genes was predicted as 100% by the Ampliseq primer design tools. As a consequence of this fact, in order to detect these variants, we decided to include in our protocol the Sanger sequencing of GCK exon 10, HNF1a exon 4 and 7, and SLC5a2 exon 11 using specific pairs of primers (available on request).

MODY is defined as an autosomal inherited form of diabetes mellitus caused by mutations in a single gene. In 2013, Lopez et al. described the co-inheritance of HNF1a and GCK mutation in one MODY patient [24]. In the last few years, with the advent of the application of NGS sequencing for genetic diagnosis of monogenic diabetes, further cases of MODY caused by defects in different genes involved in insulin release in response to blood glucose levels have been described [25]. In our validation experiments, three variants that we classified to the best of our knowledge as VUS were detected in two subjects with heterozygous GCK defects and in one with Wolfram syndrome diagnosis (Table 3). Due to the anonymization of the sample used for the panel validation, it is not an aim of this paper to discuss the role of the identified VUS in the clinical phenotype of the three patients. Herein, we only suppose that the difficulty to establish a genotype/phenotype correlation in patients with monogenic diabetes, or to understand the reason that patients with the same defect have frequently different clinical manifestations, may be related to the presence of unidentified variants in other genes that have not been sequenced. This happens principally when genetic diagnosis is performed by Sanger sequencing.

## 5. Conclusions

In conclusion, our panel based on the Ampliseq approach can be useful for identifying pathogenic variants in multiple genes in a short time. This could help to detect several defects in children and young adults who need to receive a genetic diagnosis necessary for optimal treatment. Sanger sequencing will be included in our analytical protocol to avoid missing frameshift variants due to dysglycaemia.

## Figures and Tables

**Figure 1 life-13-01080-f001:**
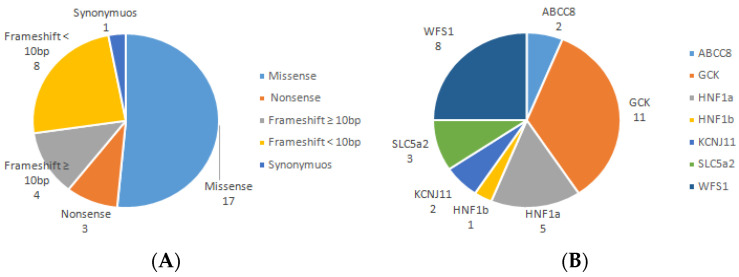
Distribution of types of mutations (**A**) and genes (**B**) used to optimize the NGS panel.

**Table 1 life-13-01080-t001:** “List of genes included in on demand panel” composed by 44 genes causative of dysglycaemia and its complications. AD—autosomal dominant, AR—autosomal recessive, XLR—X-linked recessive, ND—not determined.

Gene	Location	Phenotype	Phenotype MIM Number	Inheritance
ABCC8	11p15.1	MODY, type 12	600509	AD, AR
AIRE	21q22.3	Autoimmune polyendocrinopathy syndrome, type I, with or without reversible metaphyseal dysplasia	240300	AD, AR
ALMS1	2p13.1	Alstrom syndrome	203800	AR
APPL1	3p14.3	MODY, type 14	616511	AD
AQP2	12q13.12	Diabetes insipidus, nephrogenic, 2	125800	AD, AR
AVPR2	Xq28	Diabetes insipidus, nephrogenic, 1	304800	XLR
BBS1	11q13.2	Bardet–Biedl syndrome 1	209900	AR, DR
BLK	8p23.1	MODY, type 11	613375	AD
CISD2	4q24	Wolfram syndrome 2	604928	AR
DIAPH1	5q31.3	Deafness, autosomal dominant 1, with or without thrombocytopenia/Seizures, cortical blindness, microcephaly syndrome	124900/616632	AD/AR
GATA6	18q11.2	Pancreatic agenesis and congenital heart defects	600001	AD
GCK	7p13	MODY, type 2	125851	AD
GJB2	13q12.11	Deafness, autosomal recessive 1A	220290	AR, AD
GLIS3	9p24.2	Diabetes mellitus, neonatal, with congenital hypothyroidism	610199	AR
GLUD1	10q23.2	Hyperinsulinism-hyperammonemia syndrome	606762	AD
HADH	4q25	3-hydroxyacyl-CoA dehydrogenase deficiency/Hyperinsulinemic hypoglycemia, familial, 4	231530/609975	AR/AR
HNF1a	12q24.31	MODY, type3	600496	AD
HNF1b	17q12	MODY, type 5 Renal cysts, and diabetes syndrome	137920	AD
HNF4a	20q13.12	MODY, type 1	125850	AD
IL2RA	10p15.1	Diabetes mellitus, insulin-dependent, susceptibility to, 10/Immunodeficiency 41 with lymphoproliferation and autoimmunity	601942/606367	Nd/AR
INS	11p15.5	MODY, type 10	613370	AD
INSR	19p13.2	Hyperinsulinemic hypoglycemia, familial, 5/Leprechaunism/ Rabson–Mendenhall syndrome/Diabetes mellitus, insulin-resistant, with acanthosis nigricans	609968/246200/262190/ 610549	AD/AR/AR/Nd
KCNJ11	11p15.1	MODY, type 13	616329	AD
KFL11	2p25.1	MODY, type 7	610508	AD
LRBA	4q31.3	Immunodeficiency, common variable, 8, with autoimmunity	614700	AR
MAGEL2	15q11.2	Schaaf-Yang syndrome	615547	AD
NeuroD1	2q31.3	MODY, type 6 Maturity-onset diabetes of the young 6	606394	AD
OPA1	3q29	Mitochondrial DNA depletion syndrome 14 (encephalocardiomyopathic type)/Behr syndrome/Optic atrophy 1/Optic atrophy plus syndrome/Glaucoma, normal tension, susceptibility	616896/210000/165500/ 125250/606657	AR/AR/AD/AD/Nd
OPA3	19q13.32	3-methylglutaconic aciduria, type III/Optic atrophy 3 with cataract	258501/165300	AR/ AD
PAX4	7q32.1	MODY, type 9	612225	AD
PAX6	11p13	Coloboma of optic nerve/Coloboma, ocular/Morning glory disc anomaly/Aniridia/Anterior segment dysgenesis 5, multiple subtypes/Cataract with late-onset corneal dystrophy/Foveal hypoplasia 1/Keratitis/Optic nerve hypoplasia	120430/120200/120430/ 106210/604229/106210/136520 /148190/165550	AD/AD/AD/AD/AD/AD/AD/AD/AD
PDX1-IPF1	13q12.2	MODY, type IV/Pancreatic agenesis 1/Diabetes mellitus, type II, susceptibility	606392/260370/125853	AD/AR/AD
POU3F4	Xq21.1	Deafness, X-linked 2	304400	XLR
RFX6	6q22.1	Mitchell–Riley syndrome	615710	AR
SEL1L	14q31.1	Branchial cleft syndrome involving hypertelorism, preauricular sinus, punctal pits, and deafness	614187	AD, AR
SH2B1	16p11.2	Severe obesity, insulin resistance, and neurobehavioral abnormalities	608937	AD
SLC5A2	16p11.2	Renal glucosuria	233100	AD, AR
SOX9	17q24.3	Acampomelic campomelic dysplasia	114290	AD
SOX17	8q11.23	Vesicoureteral reflux 3	613674	AD
STAT1	2q32.2	Immunodeficiency 31A, mycobacteriosis, autosomal dominant/Immunodeficiency 31B, mycobacterial and viral infections, autosomal recessive/Immunodeficiency 31C, chronic mucocutaneous candidiasis, autosomal dominant	614892/613796/614162	AD/AR/AD
STAT3	17q21.2	Autoimmune disease, multisystem, infantile-onset, 1/Hyper-IgE recurrent infection syndrome	615952/147060	AD/AD
STAT5B	17q21.2	Growth hormone insensitivity with immune dysregulation 1, autosomal recessive/Growth hormone insensitivity with immune dysregulation 2, autosomal dominant	245590/618985	AR/AD
TMEM126A	11q14.1	Optical trophy 7	612989	AR
WFS1	4p16.1	Wolfram Syndrome 1	222300	AR

**Table 2 life-13-01080-t002:** Validation data of the “dysglycaemia panel”. ACMG—American College of Medical Genetics and Genomics, Hom—homozygous, P—pathogenic, LP—likely pathogenic, VUS—variants of uncertain significance, B—benign, Nd—not detected. VUS were described in the literature and liked to clinical phenotype.

Sample ID	Gene	Transcript	Mutation Detected by Sanger	Type of Mutation	Detected by NGS	ACMG	Additional Variants
Sample 1	WFS1	NM_006005.3	c.2389 G > A; p.Asp797Asn	Missense	Yes	P	
Sample 2	GCK	NM_000162.3	c.1279_1358delinsTTACA; p.Ser426_Ala454delinsLeuGln	Frameshift ≥ 10 bp	No	LP	PDX1: c.97C > T;p.Pro33Ser
Sample 3	GCK	NM_000162.3	c.1332_1333_dupGC; p.Ala378dup	Frameshift < 10 bp	Yes	LP	WFS1: c.2194C > T; p.Arg732Cys
Sample 4	HNF1a	NM_000545.8	c.1027_1029del2; p.Ser343fs74X	Frameshift < 10 bp	Yes	VUS	
Sample 5	SLC5a2	NM_003041.4	c.1961A > G; p.Asn654Ser	Missense	Yes	VUS [11]	
Sample 7	HNF1a	NM_000545.8	c.872dupC;p.Pro291fsinsCys	Frameshift < 10 bp	No	P	
Sample 8	KCNJ11	NM_000525.3	c.506 C > T; p.Met169Thr	Missense	Yes	P	
Sample 9	ABCC8	NM_000352.4	c.4685delC; p.Pro1563del	Frameshift < 10 bp	Yes	LP	
Sample 11	HNf1b	NM_000458.3	c.226G > T; p.Gly76Cys (rs144425830)	Missense	Yes	B	
Sample 12	GCK	NM_000162.3	c.781G > A; p.Gly261Arg	Missense	Yes	P	
Sample 13	GCK	NM_000162.3	c.1379_*2del22; p.Ala460fs	Frameshift ≥ 10 bp	Yes	LP	
Sample 14	WFS1	NM_006005.3	c.1338 C > A; p.Ser446Arg	Missense	Yes	VUS [12]	
Sample 16	WFS1	NM_006005.3	c.319 G > C; p.Gly107Arg Hom	Missense	Yes	LP	
Sample 17	ABCC8	NM_000352.4	c.916 C > T; p.Arg306Cys	Missense	Yes	VUS [13]	
Sample 18	KCNJ11	NM_000525.3	c.601C > T; p.Arg201Cys	Missense	Yes	P	
Sample 20	GCK	NM_000162.3	c.579 G > T	Synonymous	Yes	LP	
Sample 21	WFS1	NM_006005.3	c.1582 T > G; p. Tyr528Asp Hom	Missense	Yes	VUS [14]	
Sample 22	WFS1	NM_006005.3	c.2106_2113del8nt; p.V644fs64X Hom	Frameshift < 10 bp	Yes	LP	HNF1a: c.481G > A; p.Ala161Thr
Sample 23	WFS1	NM_006005.3	c.1523 A > G; p.Tyr508Cys Hom	Missense	Yes	LP	
Sample 25	HNF1a	NM_000545.8	c.262delG; p.Glu88fs	Frameshift < 10 bp	Yes	LP	
Sample 26	SLC5a2	NM_003041.4	c.1261_1280dup19; p.Glu421_Arg427dup	Frameshift ≥ 10 bp	No	LP	
Sample 27	GCK	NM_000162.3	c.214G > A; p.Gly72Arg	Missense	c.214G > A; p.Gly72Arg	P	
Sample 28	WFS1	NM_006005.3	c.1514G > A; p.Gys505Tyr c.1620_1622delGTG; p.Trp540_Cys541	Missense Frameshift < 10 bp	c.1514G > A; p.Gys505Tyr c.1620_1622delGTG; p.Trp540_Cys541	LP/ VUS [15]	
Sample 32	GCK	NM_000162.3	c.1234T > C; p.Ser412Pro	Missense	c.1234T > C; p.Ser412Pro	P	
Sample 33	HNF1a	NM_000545.8	c.1342_1374del, p.Val448_Thr458del	Frameshift ≥ 10 bp	Nd	VUS	
Sample 34	SLC5a2	NM_003041.4	c.1566C > A; p.Cys522* Hom	Nonsense	c.1566C > A; p.Cys522* Hom	LP	
Sample 35	GCK	NM_000162.3	c.490delC; p.Leu164Phefs*40	Frameshift < 10 bp	c.490delC; p.Leu164Phefs*40	P	
Sample 36	WFS1	NM_006005.3	c.1558C > T; p.Gln520* Het	Nonsense	c.1558C > T; p.Gln520* Het	P	
Sample 40	GCK	NM_000162.3	c.1302C > A; p.Cys434*	Nonsense	c.1302C > A; p.Cys434*	P	
Sample 44	HNF1a	NM_000545.8	c.1544C > T; p.Thr515Met	Missense	c.1544C > T; p.Thr515Met	VUS [16]	
Sample 45	GCK	NM_000162.3	c.1174C > T; p.Arg392Cys	Missense	c.1174C > T; p.Arg392Cys	P	
Sample 46	GCK	NM_000162.3	c.1352T > C; p.Leu451Pro	Missense	c.1544C > T; p.Thr515Met	LP	

**Table 3 life-13-01080-t003:** Analytical validation of the “dysglycaemia panel”.

Performance Metric	Value (%)	Approach
Clinical sensitivity Missense	100	17/17 detected
Clinical sensitivity Nonsense	100	3/3 detected
Clinical sensitivity Frameshift ≥ 10 bp	25	1/4 detected
Clinical sensitivity Frameshift < 10 bp	87.5	7/8 detected
Synonymous	100	1/1 detected

**Table 4 life-13-01080-t004:** Analysis of the three variants in addition to the variants previously identified by Sanger Sequencing. Het—heterozygous, ACMG—American College of Medical Genetics and Genomics, Mut Taster—MutationTaster, HGMD—Human Gene Mutation Database, VUS—variants of uncertain significance, LP—likely pathogenic, Poss D—possibly damaging, Prob D—probably damaging, D—damaging, Ref—references.

Patient Code	Gene	Transcript	Variants	Status	ACMG	Clin Var	dbSNP	gnomAD	Polyphen	Sift	Mut Taster	HGMD	Ref
Sample 2	PDX1	NM_000209.4	c.97 C > T; p.Pro33Ser	Het	VUS	VUS	rs192902098	0.00005625	Poss D	D	D	No	No
Sample 3	WFS1	NM_006005.3	c.2194C > T; p.Arg732Cys	Het	LP	VUS	rs71526458	0.00006092	Prob D	D	D	No	No
Sample 22	HNF1a	NM_000545.8	c.481G > A; p.Ala161Thr	Het	VUS	VUS	rs201095611	0.00009549	Poss D	D	D	CM981897	Chevre (1998) Diabetologia 41, 1017

## Data Availability

Not applicable.

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
