# Peer review of "Next Generation Sequencing (NGS) Target Approach for Undiagnosed Dysglycaemia"

_life, 2023, doi:10.3390/life13051080_

Round 1

Reviewer 1 Report

This paper is a study of 33 MODY patients using PGM NGS Sequencer which the mutations detected by  Sanger Sequencing.

The advantage of analysis using the NGS panel compared to Sanger is that all candidate genes can be analyzed at once, and previously unknown VUS can be confirmed.

Most mutations were detected by PGM sequencing, but frameshift mutations proved difficult to detect. However, the Discussion properly presents a plan to solve the problem.

In conclusion, although this study lacks new findings, it is considered to contain contents suitable for a brief report.

Minor Comments.

1. Each table is too big and uncomfortable to see. Needs to be updated in a better looking format.

2. I don't think there are enough legends or footnotes for each table. It would be nice if you could add a description of the data inside the table so that we can better understand the table.

The English quality is well written. 

Author Response

  1. Each table is too big and uncomfortable to see. Needs to be updated in a better looking format.

R: Table format was improved and updated as requested

  1. I don't think there are enough legends or footnotes for each table. It would be nice if you could add a description of the data inside the table so that we can better understand the table.

R: Legend and footnotes were improved as requested.

Reviewer 2 Report

In their manuscript, “Next Generation Sequencing (NGS) Target approach for undiagnosed dysglycaemia,” Aloi and colleagues present a validation study of a 44 gene NGS panel for individuals with suspected genetic forms of dysglycemia. They report that their panel achieved over 200x coverage and identified 96.5% (29/33) of the variants assessed, with four frameshift variants missed. They conclude that their approach has high technical validity for genetic diagnosis of dysglycemia and that Sanger sequencing should be incorporated into this NGS-based protocol to avoid missing frameshift variants. 

Overall, the manuscript is clearly organized; however, there are many typographical errors and misspelled words (e.g., ‘frameshit’ in paragraph 1 of the Discussion, Table 1: ‘Opticatrophy’). The topic of genetic testing for dyslipidemia is of potential interest to the greater medical genetics community, but there have already been primary studies (e.g., PMID: 34789499, PMID: 26599467) as well as in-depth reviews (e.g., PMID: 32050823); thus, the originality of this manuscript and what new insight it contributes are unclear. Importantly, detailed technical standards for NGS-based testing have been previously published (see PMID: 33927380); these are not referenced in the manuscript, such that the reader does not have the necessary context to understand the viability of the NGS panel presented for the diagnosis of suspected genetic forms of dysglycemia. Moreover, the manuscript fails to provide a detailed rationale for why these 44 genes were chosen, nor gives ACMG classifications for the variants in Table 2 and/or supporting classification criteria for the variants of uncertain significance in Table 4. The authors need to address the above issues in order for the manuscript to be suitable for publication.

As noted in my comments to the authors, the quality of the English language is poor, with many typographical errors and misspelled words (e.g., ‘frameshit’ in paragraph 1 of the Discussion, Table 1: ‘Opticatrophy’).

Author Response

R:

- typographical errors and misspelled words (e.g., ‘frameshit’ in paragraph 1 of the Discussion, Table 1: ‘Opticatrophy’) were corrected as requested

- The manuscript suggested by the referee were added in the main text as references.

- Dysglycaemia panel is composed by 44 genes. They were selected according to recently published data that describe the NGS approach for genetic diagnosis of monogenic diabetes. This sentence was added in the discussion paragraph.

- ACMG classification of the variants reported in table 2 was updated as requested

- Genetic test allowed us to confirm the presence of the knowns mutations and in three cases further variants were found in other genes included in the panel. For NGS panel validation we used anonymized DNA sample as mutated controls. Due to this fact clinical history, metabolic and biochemical data of these patients were not available and consequently establish genotype/phenotype correlation or demonstrate if a clinical symptom (i.e. hyperglicamia) has worsened is impossible in this study.

Comments on the Quality of English Language

English mother tongue revision had been done as requested

Round 2

Reviewer 2 Report

I thank Aloi and colleagues for their revisions. The manuscript is substantially improved and suitable for publication. 

The quality of the English is substantially improved